# Ultrafast non-excitonic valley Hall effect in MoS$_2$/WTe$_2$ heterobilayers

Jekwan Lee[1,2,5], Wonhyeok Heo[2,5], Myungjun Cha[1,2], Kenji Watanabe [3], Takashi Taniguchi[3], Jehyun Kim[1], Soonyoung Cha[4], Dohun Kim [1], Moon-Ho Jo [4] & Hyunyong Choi [1,2✉]

The valley Hall effect (VHE) in two-dimensional (2D) van der Waals (vdW) crystals is a promising approach to study the valley pseudospin. Most experiments so far have used bound electron-hole pairs (excitons) through local photoexcitation. However, the valley depolarization of such excitons is fast, so that several challenges remain to be resolved. We address this issue by exploiting a unipolar VHE using a heterobilayer made of monolayer MoS$_2$/WTe$_2$ to exhibit a long valley-polarized lifetime due to the absence of electron-hole exchange interaction. The unipolar VHE is manifested by reduced photoluminescence at the MoS$_2$ A exciton energy. Furthermore, we provide quantitative information on the time-dependent valley Hall dynamics by performing the spatially-resolved ultrafast Kerr-rotation microscopy; we find that the valley-polarized electrons persist for more than 4 nanoseconds and the valley Hall mobility exceeds $4.49 \times 10^3$ cm$^2$/Vs, which is orders of magnitude larger than previous reports.

[1] Department of Physics and Astronomy, Seoul National University, Seoul, Korea. [2] Institute of Applied Physics, Seoul National University, Seoul, Korea. [3] Advanced Materials Laboratory, National Institute for Materials Science, Tsukuba, Japan. [4] Center for Artificial Low Dimensional Electronic Systems, Institute for Basic Science, Pohang, Korea. [5] These authors contributed equally: Jekwan Lee, Wonhyeok Heo. ✉email: hy.choi@snu.ac.kr

The broken inversion symmetry of monolayer transition metal dichalcogenides (TMDs) leads to the valley-contrasting Berry curvature distribution that generates a topological Hall current whose signs depend on the valley index[1–3]. This VHE was first predicted in inversion symmetry broken graphene[4] and was observed in monolayer $MoS_2$[5], gapped[6], and bilayer graphene[7]. Owing to the topological nature, the valley Hall transport is expected to be free from the Joule losses[3], and also promise new logic functionalities operating in momentum space[8]. Experimentally, the finite orbital moment readily facilitates an optical means to activate the VHE by using circularly polarized photons ($\sigma^+$ or $\sigma^-$) near the TMD bandgap[5,9–11]. As such, the direct optical excitation has been one routine and yet convenient way of accessing the VHE and the associated transport phenomena.

However, the optically generated valley excitons are short-lived, typically in tens of picosecond (ps)[12,13], which pose great challenges in valleytronic applications. Microscopically, the intervalley excitonic exchange interaction significantly subdues the valley polarization relaxation time[12,13]. To overcome these issues, recent works proposed the use of type-II van der Waals heterostructures in which non-excitonic unipolar carriers, i.e., either electrons or holes, are used[14,15]. By suppressing the Maialle–Silva–Sham exchange pathway, the valley pseudospin is observed to have a much longer lifetime. Besides, the electrically-driven VHE without light excitation was also demonstrated[16], but the origin of the VHE has not been clearly explained.

Here, we report on an approach to demonstrate the optically driven non-excitonic VHE. Our approach is to inject the spin-polarized electrons in a partially stacked vertical heterobilayer that consists of a 2D monolayer of $MoS_2$ and $WTe_2$. We observed a long-lived VHE within a few nanoseconds (ns) by performing the temporally- and spatially resolved photoluminescence (PL) and Kerr-rotation microscopy. In our structure, the dynamic character of the VHE is solely driven by electrons with about 20-fold increase of valley Hall mobility compared to the exciton-based VHE[5,10,14]. Furthermore, we show that the intrinsic contribution of our unipolar VHE is deeply rooted in Berry curvature of the occupied conduction electrons, whereby we obtained a good agreement on the gate-dependent depolarization time between the theoretical estimation and the ultrafast Kerr-rotation data.

## Results

Figure 1a and b shows the schematics of our experiment and an optical microscope image of a device, respectively (see Supplementary Note 1 for the detailed device fabrication). The energy band diagram is shown in Fig. 1c. Because the 2D topological insulator (TI) supports helical electron channels with an out-of-plane spin orientation[17–20], the 1T′-phase monolayer $WTe_2$ acts as a spin-polarized unipolar electron source. The transverse VHE was measured by applying a longitudinal electric field across $MoS_2$. Electrons with a specific spin orientation were prepared at the junction between $MoS_2$ and edge of $WTe_2$ by circularly polarized 50-fs pulses with a photon energy $\varepsilon_1$ of 1.55 eV. We tuned the Fermi level to be located within the $MoS_2$ bandgap ($\Delta$) such that the optical excitation by the ultra-short pump pulse takes place only in $WTe_2$, not in $MoS_2$, i.e., $\rho < \varepsilon_1 < \Delta$; since the short-pulse pump excitation is spatially separated from the valley Hall transport region, we call it a remote pump.

Figure 1d is the schematic diagram showing the difference between the excitonic and the non-excitonic VHE. Compared to the excitonic VHE, which involves the generation and dissociation of excitons[5] or the transverse transport of exciton itself[21], our non-excitonic VHE is based on the unipolar charge transport (thus termed non-excitonic), and the transverse valley Hall occurs

in the $MoS_2$ layer with electrons only. The non-excitonic VHE in $MoS_2$ was investigated by two independent measurements. First, the transverse VHE was confirmed by spatially resolved PL measurements. Second, we performed the time-resolved Kerr-rotation (TRKR) microscopy with varying the Fermi level of the heterobilayer. While the former provides the conventional transverse Hall characteristics in a static limit[10–15], the latter enables to probe the dynamic features that were previously uninvestigated.

Figure 2a shows the results of differential PL 2D scanning using a 2.33 eV-probe diode laser measured at the A exciton resonance (~1.9 eV) of $MoS_2$ when the 50-fs, 1.55 eV "remote pump" pulse (left panel: $\sigma^+$ and right panel: $\sigma^-$) excites at a position $(x, y)$ of $(0, -0.2)$ μm. Referring to Fig. 1c, the remote pump at $WTe_2$ with energy $\varepsilon_1$ exclusively excites the spin-polarized electrons in $WTe_2$, and the electrons easily transferred to $MoS_2$. The valley index of the transferred electrons in $MoS_2$ corresponds to the locked spin-valley index, accordingly. These electrons then contribute to the differential changes of PL in the $MoS_2$ layer (Supplementary Note 4), which are measured by the local laser excitation using the 2.33 eV diode. Figure 2a clearly shows that PL signal is spatially bent in an opposite transverse direction when switching the light helicity of the remote pump. As displayed in Fig. 2b, the examination of PL spectrum shows that the A exciton peak is seemingly reduced upon the remote pump excitation; the PL spectra are measured at $(0.25, 0.1)$ μm, which is far away from the heterojunction with the remote pump $\sigma^-$ excitation. This spectral feature can be understood as following. Intuitively, if no electrons are supplied from the photoexcited $MoS_2/WTe_2$ edge, then the PL spectrum in the $MoS_2$ region should be featureless irrespective of the remote pump excitation. A significantly reshaped PL spectrum with a reduced intensity is clearly seen near 1.9 eV (A exciton energy of $MoS_2$), indicating the electrons are transferred to the $MoS_2$ layer. The result of decreased exciton population appears as a suppressed A exciton PL intensity[22]. This phenomenon is similarly observed for different gate voltages (Supplementary Note 4 and Supplementary Fig. 6). Here we note that although the remote pump area of the interior bulk is much larger than the 1D edge, the majority of spin-polarized electrons would originate from the edge with a small bulk background. This is consistent with the refs. [19,20], where the bulk is lack of distinct helical states and the density of states of the edge is significantly larger than the bulk.

Although the above experiments present the photon helicity-dependent spatial shift of electron wavepackets, but it lacks the electron spin-polarization information. To provide such information, we performed the temporally- and spatially resolved Kerr-rotation spectroscopy. Figure 3 presents the scanning TRKR data in the $MoS_2$ layer to visualize the transient Kerr signals after ultra-short remote photoexcitation in the $WTe_2$ edge. We record the 2D maps of the Kerr-rotation angle $\theta$ at different temporal delay $\Delta t$ between the remote pump and the Kerr-rotation probe pulse. Figure 3a shows the 2D maps of $\theta$ when the helicity of the remote pump is $\sigma^+$ (extended datasets are shown in Supplementary Figs. 7 and 8). In Fig. 3b, we show measurements for the Kerr angles at different $\Delta t$ obtained by a line cut at $y = 0.7$ μm. The time-dependent spatial shift of the Kerr angles corresponds to the transverse Hall transport of spin-polarized electrons. Assuming that the thermal gradient as well as the electric field of the remote pump can be approximated as a Gaussian profile, we obtain the valley Hall velocity $v_{Hall}$ of about $1.87 \times 10^3$ m/s and the corresponding Hall mobility $\mu_{Hall}$ of $4.49 \times 10^3$ cm²/Vs (Supplementary Note 6). This transverse mobility is by far larger than the best known longitudinal Hall mobility $\mu_1$ of 1020 cm²/Vs with 1D graphene edge contact[23]. This distinct difference implies that the unipolar valley Hall transport is strongly immune to the extrinsic scatterings.

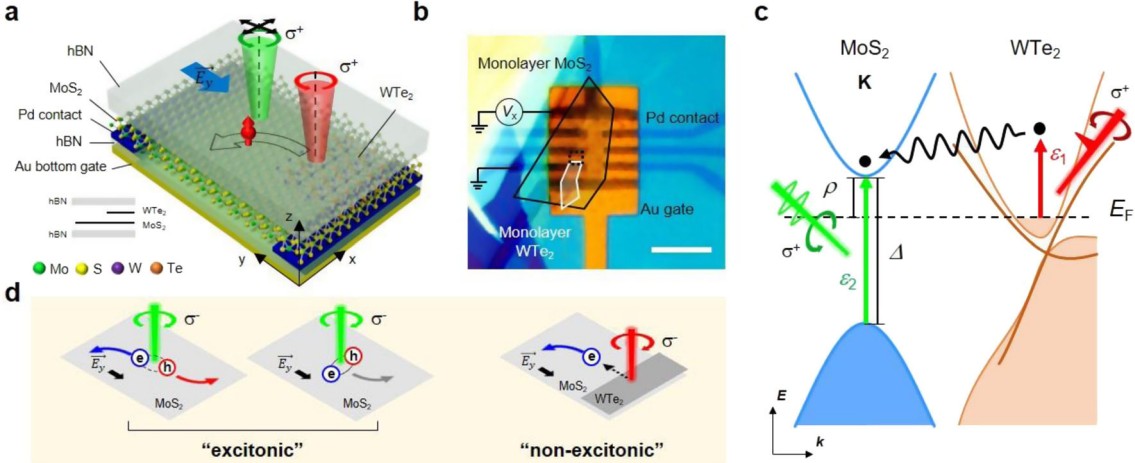

**Fig. 1 Experimental schemes of non-excitonic valley Hall effect. a** The schematic diagram of the heterobilayer device with Au bottom gate (dark yellow layer) and the remote optical measurements is shown (drawn by authors). Under the longitudinal electrical bias $\vec{E_y}$ between Pd electrodes (dark blue), the spin-polarized electrons are selectively excited by the circularly polarized ($\sigma^+$ in the figure) 50 fs 1.55 eV short pulse (red beam), and the VHE (black arrow) is observed by another laser (green beam). **b** An optical microscope image of the heterobilayer device is shown. The monolayer $MoS_2$ and $WTe_2$ are marked with the black and the white line, respectively. The dashed black line shows the spatial region where the 2D scanning experiments were performed. The longitudinal field is applied by the bias voltage $V_x$. The scale bar (white) is 10 μm. **c** A schematic energy diagram of the $MoS_2$ (blue)/$WTe_2$ (orange) heterobilayer. The short-pulse light excitation with energy $\varepsilon_1$ (red) below the bandgap of $MoS_2$ ($\Delta$) selectively excites the spin-polarized electrons (black dots) in $WTe_2$. Another light excitation with energy $\varepsilon_2$ (green) is applied to observe the effect of the injected electrons. **d** The excitonic VHE involves both the unbound (left) and the bound (right) electron–hole pairs. The non-excitonic VHE is unipolar without exciton formation or dissociation. All measurements were performed at a liquid nitrogen temperature of 78 K.

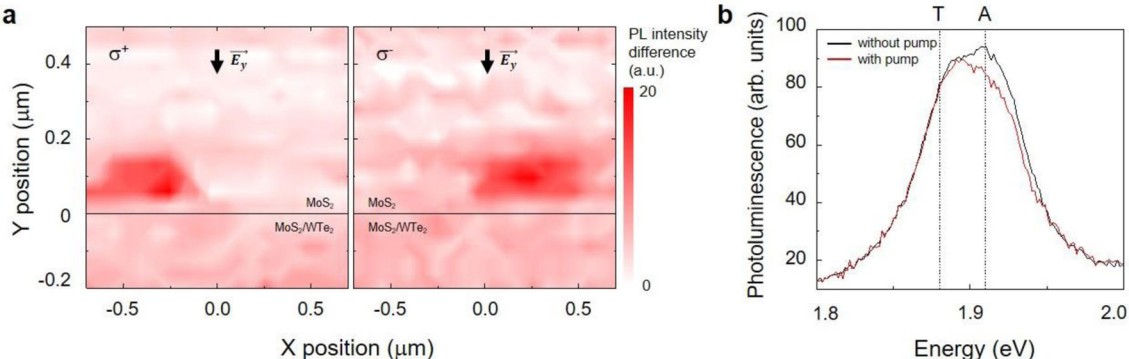

**Fig. 2 Scanning differential PL measurements across the $MoS_2$/$WTe_2$ edge boundary.** The A exciton PL signal of $MoS_2$ is suppressed by the electron transfer from $WTe_2$. **a** The spatially resolved 2D PL intensity of the $MoS_2$ A exciton ($V_G = 2$ V) is shown for the remote pump with $\sigma^+$ polarization (left) and $\sigma^-$ polarization (right). The center of the remote pump excitation with the 2 μm waist is fixed at $(0, -0.2)$ μm. The edge of the $WTe_2$ monolayer is located at the solid black line ($y = 0$ μm) and the longitudinal electrical bias $\vec{E_y}$ is applied in the direction of the black arrow. **b** PL spectrum at $(0.25, 0.1)$ μm, with (red) and without (black) the pump pulse is shown. T and A stand for the trion and the A exciton of $MoS_2$, respectively.

Having investigated the spatially-dependent valley Hall transients, we next probe more details on the temporal dynamics at a fixed probe position of $MoS_2$. We take the TRKR signals at a position of $(-0.5, 0.5)$ μm, which is again sufficiently away from the remote pump position. A representative example is shown in Fig. 3c for $V_G = 2$ V. After the TRKR signal increases with a rising time of about 37 ps, the signal reaches a peak at ~100 ps. Then a very slow decaying dynamics was observed with a time constant of about 4 ns. We exclude some possibilities involving intralayer and interlayer excitons for the long decaying transient, as discussed in the following. First, prior local TRKR investigations in monolayer $WSe_2$[24–26] found that the depolarization rate of intralayer valley polarization is around 0.5 ps$^{-1}$. Ultrafast four-wave-mixing[12] and pump-probe[27] spectroscopy have evidenced that the intervalley electron–hole exchange interaction is the origin of the rapid valley depolarization that occurs within 1 ps.

Second, the interlayer exciton dynamics[28–31] cannot contribute to our TRKR signal considering the spatially separated pump and probe, the energy band alignment of $MoS_2$ and $WTe_2$, and the existence of the bias voltage. Last, we can also exclude the relatively long spin-flip transition, simply because no sign change was observed in our long-lasting TRKR trace. Given that our valley Hall transients are dominated by electrons, we can consider such a ns-long valley lifetime as the characteristics of the conduction band split by the spin–orbit coupling (SOC), similar to the long-lived valley holes in the type-II heterostructures[15]. The small contribution of the electron–hole exchange interaction, as well as the suppressed intervalley spin-flip and the momentum transfer of electrons, are the key contributors to our long-lasting valley polarization.

The above bandstructure consideration can be substantiated by the valley-contrasting Berry curvature of conduction electron

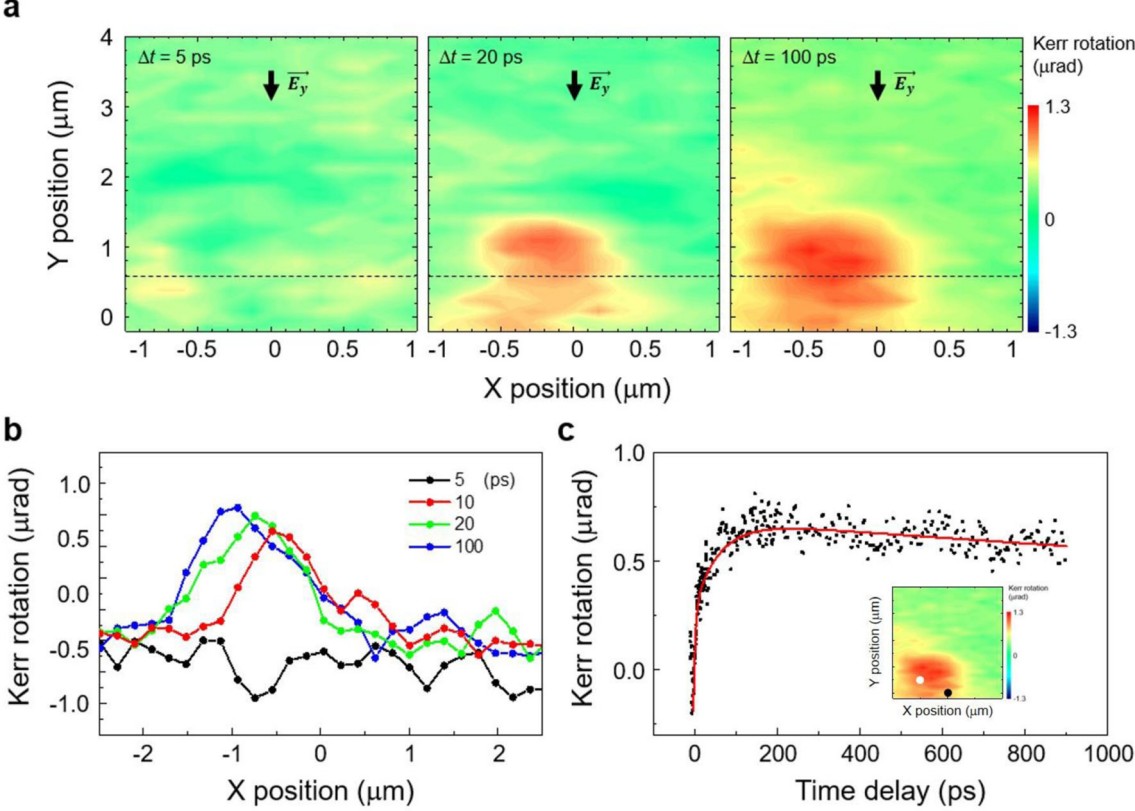

**Fig. 3 Experiment data from the TRKR microscopy. a** Time-resolved dynamics of the Kerr-rotation angle show the spatial shift of the valley-polarized electron wavepackets with increasing pump-probe delay $\Delta t$ at $V_G = 2$ V. In the experiment, the center of the remote pump was excited at a fixed position of (0, 0) and the probe was scanned the designated area at each $\Delta t$. The dashed line is a line cut for the time-dependent measurement. The corresponding data are shown in (**b**). **b** The one-dimensional line cut measurements of Kerr-rotation angles are displayed for several pump-probe delays. **c** Measured time-resolved Kerr-rotation angle (black dots) shows a long decaying transient. Solid red line shows a fitting curve of the rising and decaying of Kerr-rotation signal. Inset: the measurement was done when $V_G = 2$ V, with the center of the remote pump is at (0, 0) μm and the center of the probe is at $(-0.5, 0.5)$ μm as denoted by a black dot and white dot on the scanning result, respectively.

wavepackets. Indeed, the VHE is a process induced by the Berry curvature of the electron Bloch wavefunction. The intrinsic valley Hall velocity $\mathbf{v}_\perp$ depends on the Berry curvature and the electric field $\mathbf{E}$ applied to the electron in the band[3,32],

$$\mathbf{v}_\perp = \frac{1}{\hbar}\Omega(\mathbf{k}) \times e\mathbf{E}. \qquad (1)$$

Note the Berry curvature is an intrinsic character determined by the lattice symmetry of crystal solids. In our experiment, changing $V_G$ controls the valley Hall velocity by controlling the electron population in the K and K′ point of the $MoS_2$ conduction band, where the non-zero Berry curvature is locally concentrated (see Supplementary Note 8 for detailed explanation).

To corroborate the above rationales, we have performed the $V_G$-dependent TRKR. Figure 4a presents the TRKR results measured at $V_G$ from −1.5 to 4 V. The emerging transients can be understood as the valley-polarized electron wavepacket approaching toward the probe position due to the onset of the VHE. Therefore, the valley Hall transport velocity can be obtained from the rising time after deconvolving the initial Gaussian wavepacket. Figure 4b summarized that the rising time becomes faster, thus the Hall conductivity is larger with increasing $V_G$.

## Discussion

Based on the above interpretation, we compare the experimental data with the theoretical estimation by calculating the valley Hall conductivity (see Supplementary Note 7 for detailed calculations).

The agreement is excellent for $V_G = 1.5$–4 V (Fig. 4b). In this gate sweep regime, the Fermi level lies below the conduction band edge of $MoS_2$; the anomaly seen at $V_G = 6$ V may arise from the intraband electron excitation in the degenerately doped $MoS_2$. Note that Eq. (1) does not contain any extrinsic contributions of the VHE, such as skew-scattering and side-jump mechanisms[33]. This implies that our non-excitonic unipolar VHE is intrinsic and dominated by the Berry curvature effect. For the quantitative comparison, we have listed the valley Hall mobility $\mu_{Hall}$ and longitudinal mobility $\mu_1$ in Supplementary Table 1. Indeed, our measurement is exceptionally larger than the exciton-based VHE.

After the emerging dynamics of the valley-polarized electron wavepackets, we see that the electron wavepackets decay with a time constant of about a few ns, which we attribute to the slow valley depolarization. Two dynamic processes are associated with the corresponding origins: the valley depolarization without losing the electron population and with losing through inelastic electron scattering. Under the unipolar carrier regime of our experiment, the effect of electron–hole annihilation through either intravalley or intervalley exciton recombination may not significantly reduce the valley-depolarization lifetime. As discussed, the efficient intervalley electron–hole exchange interaction cannot cause the observed long valley-depolarization lifetime[11,34]. Figure 4c summarizes the TRKR decaying time constants. Compared to Fig. 4b, while the valley Hall velocity follows $v_{VH} \propto e^{-V_G}$, the ns-long decaying time monotonically increases, which is independent of the velocity of electron

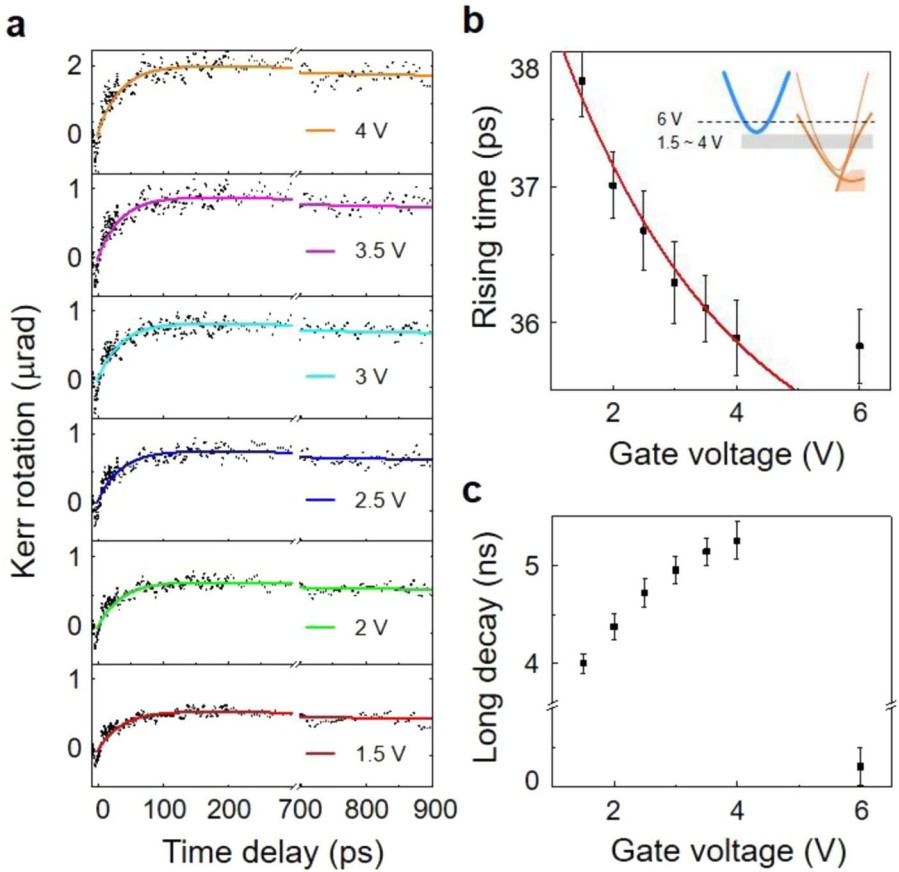

**Fig. 4 Gate-dependent Kerr-rotation dynamics. a** Time-resolved Kerr-rotation dynamics are shown with increasing $V_G$ from −1.5 to 4 V. Dots are the measured data, and the solid lines are the fitting curves. **b** The $V_G$ dependence of the rising time constant acquired from fittings in (**a**) (black dot with error bar). The obtained time constant is compared with the theoretical expectation (red line, not a fitting curve) computed from the valley Hall conductivity (Eq. (1)). Inset: the schematic energy diagram when the $V_G = 1.5$–4 V (shaded area), and $V_G = 6$ V (dashed line). **c** The $V_G$ dependence of the long decay time constant given by the fitting in (**a**). The error bars indicate standard error.

wavepackets. It represents that the intrinsic mechanism is the dominant origin in our case.

In fact, the spin splitting of the monolayer $MoS_2$ conduction band is much smaller than the valence band[35]. If the valley polarization is not originated from the injection of spin-polarized electrons due to the small spin splitting, our experiment results may purely arise from the spin-polarized electron regardless of the valley degree of freedom. However, the transverse Hall transport observed in our experiments is a strong signature of the valley Hall effect rather than the spin Hall effect, due to the following reasons. First, the Kerr-rotation signal exhibit ns-long decaying transients, denoting a far slower dynamic process when compared with the spin lifetime of the $MoS_2$ conduction band at 77 K[36], because of the highly efficient Elliot–Yafet spin relaxation of the small spin splitting. Therefore, the long-lasting Kerr-rotation signal represents the valley polarization. Second, the anomalous Hall conductivity calculated from the experiment results matches very well with the valley Hall conductivity. For an electron-doped system (such as our case), the spin Hall conductivity is about $\lambda/\Delta$ of the valley Hall effect, where $2\lambda$ is valance band spin splitting and $\Delta$ is the bandgap[1]. Considering an order-of-magnitude difference between $\lambda$ and $\Delta$ of the monolayer $MoS_2$, the valley Hall effect dominates the spin Hall effect in the total anomalous Hall transport.

In conclusion, we demonstrated a unipolar non-excitonic VHE using a $MoS_2$/$WTe_2$ heterobilayer structure. The 2D time-resolved Kerr-rotation microscopy strongly supports that the

transverse Hall dynamics is governed by the intrinsic Berry curvature effect. Although this establishes a demonstration of non-excitonic and purely intrinsic VHE, it reserves a much room to improve for the practical 'valleytronics'. For example, the use of large SOC-coupled valence bands shall increase the valley-polarized lifetime by a few orders of magnitude. Such a perspective potential, together with the unique TMD/TI heterobilayer characteristics, can pave a way toward a new 2D platform for realizing valley-based information processing.

## Methods
The detailed information about the device fabrication and the experimental setup are available in the Supplementary Information.

## Data availability
All relevant data are reported in the main text and the associated Supplementary Information. All data are available from the corresponding author upon reasonable request.

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

## Acknowledgements

J.L., W.H., M.C., and H.C. were supported by the National Research Foundation of Korea (NRF) through the government of Korea (MSIP) (Grant NRF-2018R1A2A1A05079060, NRF-2020M3F3A2A03082472), Creative Materials Discovery Program (Grant 2017M3D1A1040828), Scalable Quantum Computer Technology Platform Center (Grant 2019R1A5A1027055), and the Institute for Basic Science (IBS), Korea under Project code IBS-R014-G1-2018-A1. Part of this study has been performed using facilities at IBS Center for Correlated Electron Systems, Seoul National University.

## Author contributions

J.L. and W.H. contributed equally to this work. H.C and J.L. conceived the main idea, designed the experimental protocols, and performed preliminary experiments. W.H. and M.C. performed sample fabrication. W.H. performed the major experimental tasks with the support of J.L., K.W., and T.T. provided high-quality hBN single crystal. J.K. and D.K. measured the basic characteristics of devices. S.C. and M.-H.J. performed the measurement of basic properties for single crystals. H.C. supervised the project. J.L., W.H., and H.C. wrote the manuscript with input from all authors.

## Competing interests

The authors declare no competing interests.
