## [Peer Review File · Nature Communications]

Reviewers' Comments:

Reviewer #1:

Remarks to the Author:

The paper explores the valley hall effect for electrons in MoS₂ by using an optical injection of electrons from a WTe₂ layer. The authors use time and space resolved Kerr rotation to estimate the valley polarization lifetime and mobility of these optically injected electrons. Overall, I find the paper interesting, and the results appear reasonable. I would like to see the Kerr rotation data for both pump helicities in Figure 3, since the sign of the Kerr rotation signal should flip sign. It is somewhat surprising to me that the optical injection scheme would work reliably. What is the estimated valley polarization injection efficiency? Data from another device would be useful to compare.

Reviewer #2:

Remarks to the Author:

The manuscript by Lee et al. reports unipolar valley Hall effect using optical pumping of MoS₂/WTe₂ heterobilayer. The basic idea is generating spin-polarized electrons in WTe₂ by helical light and then transferring to MoS₂ conduction band with the same spin-valley index. The valley Hall effect was then examined by spatial and temporal resolved Kerr rotation measurements at the monolayer MoS₂ area, where a spatial shift of electron drift under a longitudinal electrical bias was observed. The study is interesting, providing a feasible route to realize injections of spin polarized electrons into MoS₂ for future applications of spintronics and valleytronics based on monolayer semiconductors. Therefore, this manuscript deserves publications.

However, the authors should address the following comments before publications.

1. The spin-polarized electron transfer from WTe₂ to MoS₂ was illustrated by spatial mapping of MoS₂ PL changes near the edge of heterojunction under laser excitation with different helicity. To demonstrate the helicity dependence, the author should always show not only the sigma plus and sigma minus polarizations, but also linear polarization to illustrate the results without helicity dependence. Furthermore, spatial mapping of PL changes without applying the longitudinal electrical bias should be demonstrated, at least in the supplementary information.
2. Did the authors check the valley polarization of trion PL in MoS₂ under circularly polarized optical pumping at WTe₂? This can give the reader some ideas about whether the injection of spin polarized electrons affect the preferential spin configuration of excitons and trions in MoS₂.
3. In spatial resolved Kerr measurements, the authors used a "non-local pump" at a specific position and then probe was then scan through the area of interest. The "non-local pump" is very confusing. It is actually a local pump at a specific position. The authors should use a more suitable terminology.
4. The rise time of Kerr signal is explained as the transport of valley-polarized electron packet toward the probe position. To verify this, the authors should demonstrate at least another probe position at a different distance.
5. The authors used gate dependent rise time of Kerr signal to demonstrate the change of Hall mobility with the Fermi energy (the occupied electron states). However, gating the MoS₂ only increase the nonpolarized electrons in MoS₂. How can it achieve a higher Hall mobility? To increase the spin-polarized electrons, increasing the excitation power at WTe₂ would make more sense.
6. How can the authors verify that the spin-polarized electrons are generated from the 1D helical edge of 2D WTe₂, rather than the interior area of WTe₂?

Reviewer #3:

Remarks to the Author:

This manuscript reports the valley Hall effect of spin-polarized electrons in MoS₂/WTe₂. Experimental signatures from both the excitonic PL intensity variation and Kerr-rotation are indicated, which unambiguously mark the presence of unipolar valley Hall transport in monolayer MoS₂. The anomalous Hall effect with a topological origin in two-dimensional layered semiconductors is an attracting topic, and several experiments performed in monolayer transition metal dichalcogenides have indicated such phenomena in recent years. The valley Hall effect of electrons in this manuscript can serve as an important complement to the previous experiments related to excitons. The experimental data shown in the manuscript is beautiful and clear. However there are still several points need the authors to clarify before I can recommend the publication:

(1) The sign of the electron Berry curvature, which determines the Hall velocity direction under a given in-plane electric field, is related to the valley but not the spin. The existence of a valley Hall effect then implies that the excited spin-polarized electrons are also valley polarized. A transport measurement has indicated that in MoS₂ the conduction band spin splitting is extremely small (~ 0.8 meV, see Nat. Commun. 8, 1938 (2017)), thus it's hard to say that the electron's spin and valley indices are locked. So the authors should discuss how the spin polarization in WTe₂ is converted to valley polarization in MoS₂. Meanwhile from the experimental observation one can get the sign of the Berry curvature thus the electron valley index. Is the obtained spin-valley relation consistent with those in the other papers?

(2) More details are needed to understand the relation between the rising time and the gate voltage in FIG. 4b. The magnitude of the valley Hall velocity is given by the product of the Berry curvature and the electric field, whereas the Hall conductivity also depends on the electron density. Is the different rising time here caused only by the Berry curvature's variation with the electron energy or wave vector k (given in Supplementary Note 7)? If so, what is the relation between V_G and the Berry curvature? On the other hands, does the electron density play any role here?

Point-by-point responses to the issues raised by the reviewers

General remarks and comment of Reviewer 1:

The paper explores the valley hall effect for electrons in MoS₂ by using an optical injection of electrons from a WTe₂ layer. The authors use time and space resolved Kerr rotation to estimate the valley polarization lifetime and mobility of these optically injected electrons. Overall, I find the paper interesting, and the results appear reasonable.

Response 1 general remarks:

We appreciate the time that Reviewer 1 took to consider our manuscript. Following the advice from Reviewer 1, we provided additional supplementary information in the revised manuscript by providing more data obtained from control.

We wish to point out that we have made a device label following Reviewer 1's advice; in the revised manuscript, device # 1 means the device we used for the most of experiments described in the main text, and device #2 indicates the auxiliary device shown in Supplementary Fig. 4. This is because Reviewer 1 suggested to provide data from another device to compare the injection efficiency of spin-polarized electrons. The additional experiments performed using the above devices are supplemented in the revised manuscript. Accordingly, we have labeled the device number in the revised manuscript.

Figure R1. Optical microscope images of devices used in the experiments. **a.** Device #1 is used for most of the experiments in the main text. **b.** Device #2 and its associated properties are added in Supplementary Fig. 4 and 7.

We also wish to note that the term “non-local pump” is replaced with “remote pump” in the revised manuscript following comment from Reviewer 2.

Below we present our point-by-point response to Reviewer 1's comments.

Comments 1-1:

I would like to see the Kerr rotation data for both pump helicities in Figure 3, since the sign of the Kerr rotation signal should flip sign.

Response 1-1:

Indeed, the Kerr rotation signal has an opposite sign when the pump helicity is reversed. The experiment result with such helicity dependence is provided in Supplementary Fig. 7 (also presented below). The 2D scanning Kerr rotation data with an opposite pump helicity (compared to the pump helicity shown in Fig. 3) clearly shows a sign flip of the signal, and the valley Hall transport flows in the opposite direction (compare to the pump helicity in Fig. 3).

Figure R2. Supplementary Fig. 8. When an opposite pump helicity is used (compared to the pump used in Fig. 3), the Kerr rotation signal flips its sign.

Comments 1-2:

It is somewhat surprising to me that the optical injection scheme would work reliably. What is the estimated valley polarization injection efficiency? Data from another device would be useful to compare.

Response 1-2:

We estimate the valley polarization injection efficiency from PL and transport data of devices #1 and #2. In short, our estimation is based on the following procedure. First, the injected electron population affects the MoS₂ Fermi level, resulting in changes in the PL spectrum. Then, the injected electron population was estimated by comparing the V_G -dependent PL spectra with and without the remote pump.

Figure R3 (Supplementary Fig. 7) shows the effect of the remote pump on the PL spectrum. Qualitatively, we see that the exciton PL with a remote pump (red color line) similarly corresponds to the PL spectrum without the remote pump when V_G is just below 3 V for both devices. Based on these observations, we can draw a conclusion that the effect of the remote pump would render the MoS₂ PL spectrum to lie at most V_G of 3 V. Here, we wish to note that although we show the PL data when V_G is 2 V without the remote pump, this corresponds to 0 % injection efficiency because we measure the PL with the remote pump when V_G is 2 V. Therefore, we can only estimate the upper bound of the injection efficiency.

For the quantitative estimation of the valley polarization injection efficiency, we use the V_g -dependent transfer data (Fig. R3b for device #1 and Fig. R3d for device #2). The change of electron population was inferred by calculating the ratio between the current I_{2V} and I_{3V} when $V_G = 2$ V and 3 V, respectively. As for device #1, $I_{2V} = 0.695$ nA and $I_{3V} = 0.761$ nA so the upper bound of efficiency $(I_{3V}-I_{2V})/I_{2V}$ is 11.2 %. Similarly, for device #2, $I_{2V} = 0.876$ nA and $I_{3V} = 0.796$ nA for device #2, thereby the upper bound of efficiency is 9.8 %.

Figure R3. Experimental results for the efficiency estimation. **a.** PL spectra with the remote pump when $V_G = 2$ V compared to the PL at $V_G = 2, 3$ V without the remote pump, and **b.** V_G -dependent transfer curve of device #1. **c** and **d** show the same experimental results for device #2.

We have added the estimated efficiency of both devices to Supplementary Note 4.

Edited contents

Added sentences

- Supplementary Note4:

(Supplementary Information, page 10, line 7)

Supplementary Fig. 7 shows the effect of the remote pump on the PL spectrum (Supplementary Fig. 6c and Supplementary Fig. 1b are shown again for the better comparison.). Qualitatively, we see that the exciton PL with a remote pump (red color line) similarly corresponds to the PL spectrum without the remote pump when V_G is just below 3 V for both devices. Based on these observations, we can draw a conclusion that the effect of the remote pump would render the MoS₂ PL spectrum to

lie at most V_G of 3 V. Here, note that although the PL data when V_G is 2 V without the remote pump is presented, this corresponds to 0 % injection efficiency because the PL with the remote pump is measured when V_G is 2 V. Therefore, we can only estimate the upper bound of the injection efficiency. For the quantitative estimation of the valley polarization injection efficiency, we use the V_G -dependent transfer data (Supplementary Fig. 7b for device #1 and Supplementary Fig. 7d for device #2). The change of electron population was inferred by calculating the ratio between the current I_{2V} and I_{3V} when $V_G = 2$ V and 3 V, respectively. As for device #1, $I_{2V} = 0.695$ nA and $I_{3V} = 0.761$ nA so the upper bound of efficiency $(I_{3V}-I_{2V})/I_{2V}$ is 11.2 %. Similarly, for device #2, $I_{2V} = 0.876$ nA and $I_{3V} = 0.796$ nA for device #2, thereby the upper bound of efficiency is 9.8 %.

Edited figure

- Figures R3 is newly added as Supplementary Fig 7.

General remarks and comment of Reviewer 2:

The manuscript by Lee et al. reports unipolar valley Hall effect using optical pumping of MoS₂/WTe₂ heterobilayer. The basic idea is generating spin-polarized electrons in WTe₂ by helical light and then transferring to MoS₂ conduction band with the same spin-valley index. The valley Hall effect was then examined by spatial and temporal resolved Kerr rotation measurements at the monolayer MoS₂ area, where a spatial shift of electron drift under a longitudinal electrical bias was observed. The study is interesting, providing a feasible route to realize injections of spin polarized electrons into MoS₂ for future applications of spintronics and valleytronics based on monolayer semiconductors. Therefore, this manuscript deserves publications.

However, the authors should address the following comments before publications.

Response:

We appreciate the effort that Reviewer 2 provided us with valuable inputs. Reviewer 2 suggested to provide a control experiment of the VHE under linearly polarized light and pointed out a confusing term, namely “non-local pump”. He/she additionally raised a few concerns about the effects of trions and excitons on the spectrally resolved PL and the dynamics of Kerr-rotation transients.

We fully agree with all Reviewer 2’s comments; we have added extra data that were not presented in the previous manuscript and revised some of confusing statements. For the terminology, the term “non-local pump” is replaced with “remote pump” to prevent any confusion.

We wish to point out that we have made a device label following Reviewer 1’s advice; in the revised manuscript, device # 1 means the device we used for the most of experiments described in the main text, and device #2 indicates the auxiliary device shown in Supplementary Fig. 4. This is because Reviewer 1 suggested to provide data from another device to compare the injection efficiency of spin-polarized electrons. The additional experiments performed using the above devices are supplemented in the revised manuscript. Accordingly, we have labeled the device number in the revised manuscript.

Figure R4. Optical microscope images of devices used in the experiments. **a.** Device #1 is used for most of the experiments in the main text. **b.** Device #2 and its associated properties are added in Supplementary Fig. 4 and 7.

Below we present our point-by-point response to Reviewer 2's comments.

Comments 2-1:

1. The spin-polarized electron transfer from WTe₂ to MoS₂ was illustrated by spatial mapping of MoS₂ PL changes near the edge of heterojunction under laser excitation with different helicity. To demonstrate the helicity dependence, the author should always show not only the sigma plus and sigma minus polarizations, but also linear polarization to illustrate the results without helicity dependence. Furthermore, spatial mapping of PL changes without applying the longitudinal electrical bias should be demonstrated, at least in the supplementary information.

Response 2-1:

We agree that the VHE should include not only the circular helicity dependence but also the effect of linearly polarized light excitation. Indeed, we performed the same experiment with a linearly polarized light but have excluded these data in the original manuscript. This was simply because the data did not reveal noticeable features.

In the revised manuscript, we have added the spatially resolved PL data with an 1.55 eV linearly polarized remote pump excitation, taken from device #1. The corresponding data is presented in Fig. R5. We have observed that the suppressed exciton PL is distributed uniformly along the edge with no noticeable valley Hall deflection. This implies the spatial distribution of K and K' valley polarization is equally spread due to the balanced up and down spin-polarized electrons.

Figure R5. Spatially resolved differential PL (taken from device #1) is presented when the remote pump is linearly polarized.

For the zero longitudinal bias voltage, we note that although the VHE is not expected to appear in the spatially resolved differential PL, our data (Fig. R6) show that such differential

PL is not completely absent. Instead, we see a signature of the VHE, though the magnitude is very weak (about 20 % of differential PL when the bias voltage is applied). The transverse spatial displacement of the VHE is quite short ($\sim 0.2 \mu\text{m}$) compared to the non-zero bias voltage ($\sim 0.4 \mu\text{m}$, see Fig. 2a in the revised manuscript). Although there is no longitudinal bias voltage, the possible scenario may include the thermal gradient and thermoelectric photocurrent, which may give rise to effective potential gradient. Of course, the small magnitude of valley polarization is because the spin-polarization injection should strongly depend on the longitudinal field strength.

This newly measured dataset of Figs. R5 and 6 are added to the Supplementary Information to complete our discussion about the effect of longitudinal electric field on the VHE.

Figure R6. In the absence of the applied longitudinal electrical field, we still observe a signature of the valley Hall transport. The data were taken from device #1 when the polarization of the remote pump is **a.** σ^+ and **b.** σ^- .

Edited contents

Added sentences

- Supplementary Note 4

(Supplementary Information page 9, line 16)

Supplementary Fig. 6c shows the spatially resolved PL data with an 1.55 eV linearly polarized remote pump excitation. We have observed that the suppressed exciton PL is distributed uniformly along the edge with no noticeable valley Hall deflection. This implies the spatial distribution of K and K' valley polarization is equally spread due to the balanced up and down spin-polarized electrons. For the zero longitudinal bias voltage, we note that although the VHE is not expected to appear in the spatially resolved differential PL, our data (Supplementary Fig. 6d, e) show that such

differential PL is not completely absent. Instead, we see a signature of the VHE, though the magnitude is very weak (about 20 % of differential PL when the bias voltage is applied). The transverse spatial displacement of the VHE is quite short ($\sim 0.2 \mu\text{m}$) compared to the non-zero bias voltage ($\sim 0.4 \mu\text{m}$, see Fig. 2a in the revised manuscript). Although there is no longitudinal bias voltage, the possible scenario may include the thermal gradient and thermoelectric photocurrent, which may give rise to effective potential gradient. Of course, the small magnitude of valley polarization is because the spin-polarization injection should strongly depend on the longitudinal field strength.

Edited figures

- Figure R5 is added in the Supplementary Information as Supplementary Fig. 6c.
- Figure R6a and b are added in the Supplementary Information as Supplementary Fig. 6d and e, respectively.

Comments 2-2:

2. Did the authors check the valley polarization of trion PL in MoS₂ under circularly polarized optical pumping at WTe₂? This can give the reader some ideas about whether the injection of spin polarized electrons affect the preferential spin configuration of excitons and trions in MoS₂.

Response 2-2:

The valley polarization of exciton and trion can be investigated by exciting ‘remote pump’ in WTe₂ and by measuring the changes of PL in MoS₂; note that we inject a spatially separated ‘pump’ light in MoS₂ layer to measure the PL.

In Fig. R7 below, we plot the helicity-resolved PL spectra when σ^+ remote pump is excited in WTe₂; note that the black line represents the MoS₂ PL without the remote pump. When the spin-polarized electron is injected by the σ^+ remote pump, the increased trion PL with the decreased exciton PL appears in the σ^+ component of PL (red line). The σ^- component of PL (blue line) is nearly the same as PL without the remote pump.

Figure R7. PL spectra of MoS₂ under σ^+ remote pump excitation in WTe₂. For comparison, we show the MoS₂ PL spectrum (black line) when $V_G = 2$ V with no remote pump in WTe₂. The red and blue lines are the helicity-resolved PL spectra (σ^+ and σ^- PL, respectively) when $V_G = 2$ V under σ^+ remote pump excitation.

As having discussed in Supplementary Note 4, such PL change is induced by the interlayer charge transfer (electron population transfer from WTe₂ to MoS₂) due to the remote pump excitation in WTe₂. The injected electrons are spin-polarized such that the electrons would fill the conduction band in MoS₂ whose valley index is matched with the associated electron

spin, i.e. spin-valley locking. Here, if the electron spin experiences a rapid decoherence within the interlayer charge transfer time, the PL changes in MoS₂ should not exhibit a noticeable difference between the σ^+ and σ^- PL in MoS₂. In fact, our experiment shows that this is not the case.

The red line in Fig. R7 shows that the trion PL is enhanced while the exciton PL decreases. Of course, the amount of changes of trion PL is not as large as the exciton PL changes. This is because of the weak oscillator strength of the trion. Nevertheless, in our **Response 1-2** to Reviewer 1, the spin-polarization injection efficiency is qualitatively and quantitatively estimated; the results are 11.2 % for device #1 and 9.8 % for device #2. The newly added data of Fig. R7 implies the valley polarization of trion PL also follows the helicity of the remote pump. Because both the exciton and trion follow the same helicity, some part of the transferred electron population contributes to the suppressed exciton PL, and the others lead to the enhanced trion PL.

Edited contents

Added sentences

- Supplementary Note 4:

(Supplementary Information, page 8, line 19)

In Supplementary Fig. 5d, we plot the helicity-resolved PL spectra when σ^+ remote pump is excited in WTe₂; note that the black line represents the MoS₂ PL without the remote pump. When the spin-polarized electron is injected by the σ^+ remote pump, the increased trion PL with the decreased exciton PL appears in the σ^+ component of PL (red line). The σ^- component of PL (blue line) is nearly the same as PL without the remote pump.

As having discussed in this note, such PL change is induced by the interlayer charge transfer (electron population transfer from WTe₂ to MoS₂) due to the remote pump excitation in WTe₂. The injected electrons are spin-polarized such that the electrons would fill the conduction band in MoS₂ whose valley index is matched with the associated electron spin, i.e. spin-valley locking. Here, if the electron spin experiences a rapid decoherence within the interlayer charge transfer time, the PL changes in MoS₂ should not exhibit a noticeable difference between the σ^+ and σ^- PL in MoS₂. In fact, the experiment result shows that this is not the case.

Added figure

- Figure R7 is added to the supplementary information as Supplementary Fig. 5d.

Comments 2-3:

3. In spatial resolved Kerr measurements, the authors used a “non-local pump” at a specific position and then probe was then scan through the area of interest. The “non-local pump” is very confusing. It is actually a local pump at a specific position. The authors should use a more suitable terminology.

Response 2-3:

The reason we used the notion of “non-local pump” was to indicate that the VHE induced in the monolayer MoS₂ is caused by the spatially separated light excitation in the monolayer WTe₂. The light excitation in WTe₂ is sufficiently far away from the monolayer MoS₂. To measure the MoS₂ PL, we use another light excitation. Because there are two light excitations (one in WTe₂ and another in MoS₂), these two excitations should be distinguishable.

We understand that the “non-local” term would be confusing to the readers. In fact, the intended meaning of “non-local pump” is closer to the meaning of spatially separated (, or remote) light excitation. Therefore, we have changed the term “non-local pump” into “remote pump” to convey the proper intention. For the measurement of MoS₂ PL, we keep the term “pump” since the light excitation in MoS₂ is necessary to measure the differential PL.

Edited contents

Edited terminology

- The term “non-local pump” used in the main text and in the supplementary information is changed to “remote pump”.

Comments 2-4:

4. The rise time of Kerr signal is explained as the transport of valley-polarized electron packet toward the probe position. To verify this, the authors should demonstrate at least another probe position at a different distance.

Response 2-4:

Indeed, it is important to verify the assumption that the rising time shown in the time-resolved Kerr rotation measurement is originated from the spatially moving valley-polarized electron packet. We should have measured the Kerr dynamics at another probe position away from the pump. While the data from another probe position would provide a convincing evidence of the VHE, we wish to note that since the spatial dimension of the observable area near the WTe₂ edge (marked as dashed black line in Fig. 1 in the main text) is very similar to the probe spot size. It was extremely challenging for us to find another probe position. In fact, while preparing the revised manuscript, we have tried to measure at another spot with 0.7 μm away from the pump. The evidence was quite marginal so that a conclusive argument cannot be made. In fact, this is one of the main reasons why we took so long time before submitting the revised manuscript.

We feel sorry that we could only measure the time-resolved Kerr rotation data at one specific position (Fig. 3 c in the revised manuscript). Although the transients were measured at one specific position, we believe that we have provided clear evidence for the dynamics of the valley-polarized electron packet by measuring both spatially and temporally resolved Kerr rotation (Fig. 3 a in the main text and Supplementary Fig. 8).

Comments 2-5:

5. The authors used gate dependent rise time of Kerr signal to demonstrate the change of Hall mobility with the Fermi energy (the occupied electron states). However, gating the MoS₂ only increase the nonpolarized electrons in MoS₂. How can it achieve a higher Hall mobility? To increase the spin-polarized electrons, increasing the excitation power at WTe₂ would make more sense.

Response 2-5:

Before replying to this comment, we have realized that Reviewer 2 and Reviewer 3 raises almost the same concern (see **Response 2-5 and Response 3-2**). So, we simply address the same explanation here.

Both Reviewer 2 and 3 raised a concern about the role of V_G on the valley Hall velocity. In our experimental regime, the different rising time likely arises from the Berry curvature variation rather than the electrostatically induced uniform charge carriers.

For the valley Hall mobility, we wish to note that it does not exactly mean that mobility is proportional to the amount of injected electrons. Instead, changing V_G , i.e. increase or decrease of the total electron population, leads to the changes of valley Hall mobility by rendering more or less electrons to be affected by a larger or smaller Berry curvature distribution in the momentum space [*Phys. Rev. Lett.* **99** 236809 (2007)].

Just to illustrate the main point, we reiterate once again that the intrinsic valley Hall velocity \mathbf{v}_\perp depends on the Berry curvature $\Omega(\mathbf{k})$ and the electric field \mathbf{E} applied to the electron in the band [*Rev. Mod. Phys.* **82** 1539-1592 (2010), *Phys. Rev. B* **75** 045315 (2007)],

$$\mathbf{v}_\perp = \frac{1}{\hbar} \Omega(\mathbf{k}) \times e\mathbf{E} .$$

Here, the Berry curvature is an intrinsic character determined by the lattice symmetry of the involved atomic structure. As long as the longitudinal electric field is constant, which for most of the cases are done by applying DC source-drain potential gradient [*Science* **344** 1489-1492 (2014); *Nat. Nanotechnol.* **11** 421-425 (2016); *Nat. Commun.* **10** 611 (2019)], the transverse valley Hall velocity would not be changed. This is exactly the same as our case.

Figure R8. Schematic diagram describing the effect of the longitudinal field (\vec{E}_y) on the electron distribution. K point of the momentum space is expressed as a red dashed line, and the bottom of the conduction band near the Fermi surface E_F is marked as a black dashed line. As V_G increases, the increased amount of electron injection and the higher Fermi surface lead the change of electron distribution after thermalization from (i) to (ii). Note that Berry curvature is concentrated near at K point. More electrons experience a larger Berry curvature in case (ii) compared to the case (i).

In our experiment, changing the electron density by V_G controls the electron distribution in the MoS₂ conduction band. The applied longitudinal electric field \vec{E}_y makes the Fermi surface being tilted in momentum space, as schematically shown in Fig. R8. After the thermalization and cooling are finished, the injected group of electrons fills the conduction band from the point marked as a black dashed line in Fig. R8. When V_G increases, the electron population at the K (or K') point increases, whose effect is explained as the band filling from (i) to (ii). This appears as a faster rising dynamic component in our time-resolved Kerr rotation because the non-zero Berry curvature is concentrated at the K and K' point of the band extrema in the momentum space, i.e. the tilted Fermi surface makes the injected electrons away from the K or K' point.

We thank both Reviewer 2 and Reviewer 3 to point out this important issue. We have included the above discussion in the revised manuscript as well as in the revised Supplementary Note 8.

Edited contents

Edited sentences

- Main text:
(Main text, page 6, line 22)

The intrinsic valley Hall velocity \mathbf{v}_\perp depends on the Berry curvature and the electric field \mathbf{E} applied to the electron in the band [3,32],

$$\mathbf{v}_\perp = \frac{1}{\hbar} \Omega(\mathbf{k}) \times e\mathbf{E} . \quad (1)$$

Note the Berry curvature is an intrinsic character determined by the lattice symmetry of crystal solids. In our experiment, changing V_G controls the valley Hall velocity by controlling the electron population in the K and K' point of the MoS₂ conduction band, where the non-zero Berry curvature is locally concentrated (see Supplementary Note 8 for detailed explanation).

(Main text, page 7, line 12)

Based on the above interpretation, we compare the experimental data with the theoretical estimation by calculating the valley Hall conductivity (see Supplementary Note 7 for detailed calculations).

(Main text, page 8, line 6)

Compared to Fig. 4b, while the valley Hall velocity follows $v_{\text{VH}} \propto e^{-V_G}$, the ns-long decaying time monotonically increases, which is independent of the velocity of electron wavepacket. It represents that the intrinsic mechanism is the dominant origin in our case.

- Supplementary Note 7:

(Supplementary Information, page 13, line 13)

The valley Hall velocity v_{VH} can be expressed as equation (1) using the electric field and the Berry curvature [11]. Besides, we used the approach of calculating valley Hall conductivity σ_{VH} to estimate the valley Hall mobility.

$$\sigma_{\text{VH}} = -\xi \frac{e^2}{\hbar} \sum_n \int \frac{d\mathbf{k}}{(2\pi)^2} f(\varepsilon_n(\mathbf{k})) \Omega_c(\mathbf{k}), \quad (\text{S5})$$

where ξ is the antisymmetric tensor, n is the band index in the 2D limit, $f(\varepsilon_n(\mathbf{k}))$ is Fermi-Dirac distribution, and $\Omega_c(\mathbf{k})$ is the Berry curvature derived in equations (S3) and (S4). Based on the Berry curvature and known parameters of the monolayer MoS₂ [10], the intrinsic contribution can be calculated. Then, the valley Hall transport mobility μ_{Hall} is derived as a function of ρ for the comparison with the experimental results.

Added sentences

- Supplementary Note 8:

(Supplementary Information, page 14)

Supplementary Note 8 – Gate voltage dependence of the valley Hall velocity

For the valley Hall mobility, we wish to note that it does not exactly mean that mobility is proportional to the amount of injected electrons. Instead, changing V_G , i.e. increase or decrease of the total electron population, leads to the changes of valley Hall mobility by rendering more or less electrons to be affected by a larger or smaller Berry curvature distribution in the momentum space.

Just to illustrate the main point, we reiterate once again that the intrinsic valley Hall velocity \mathbf{v}_\perp depends on the Berry curvature $\Omega(\mathbf{k})$ and the electric field \mathbf{E} applied to the electron in the band as shown in equation (1). As long as the longitudinal electric field is constant, which for most of the cases are done by applying DC source-drain potential gradient [4,12,15], the transverse valley Hall velocity would not be changed. This is exactly the same as our case. In our experiment, changing the electron density by V_G controls the electron distribution in the MoS₂ conduction band. The applied longitudinal electric field \vec{E}_y makes the Fermi surface being tilted in momentum space, as schematically shown in Supplementary Fig. 13. After the thermalization and cooling are finished, the injected group of electrons fills the conduction band from the point marked as a black dashed line in Supplementary Fig. 13. When V_G increases, the electron population at the K (or K') point increases, whose effect is explained as the band filling from (i) to (ii). This appears as a faster rising dynamic component in our time-resolved Kerr rotation because the non-zero Berry curvature is concentrated at the K and K' point of the band extrema in the momentum space, i.e. the tilted Fermi surface makes the injected electrons away from the K or K' point.

Added figure

- Figure R8 is newly added to the supplementary information as Supplementary Fig. 13.

Comments 2-6:

6. How can the authors verify that the spin-polarized electrons are generated from the 1D helical edge of 2D WTe₂, rather than the interior area of WTe₂?

Response 2-6:

Experimentally, it is very challenging to figure out the exact origin of the spin-polarized electrons, because the remote pump generates the photo-excited carriers both in the metallic edge as well as in the interior bulk. Here, we need to consider the following two factors.

First, although the photo-excitation area of the interior bulk is much larger than the 1D edge, the interior bulk would not generate highly spin-polarized electrons, because the bulk is lack of the distinct helical states. It implies that any carriers from the bulk interior would contribute to MoS₂ PL as a small background signal for the helicity-resolved PL.

Second, the density of states of the edge is quite larger than that of the bulk. In Fig. 9, we provide experimental and theoretical investigation from published literatures. Figure R9a [*Nat. Phys.* **13** 683-687 (2017)] shows the dI/dV spectra taken across the step edge of an 1T'-WTe₂ monolayer using a scanning tunneling spectroscopy (STS). The STS result implies that the density of state is strongly confined near the 1D edge of the monolayer 1T'-WTe₂, in which the localized conductance along the sample is highly visible. Figure R9b [*Adv. Mater.* **28** 4845-4851 (2016)] is the calculated wavefunction amplitude integrated along the length direction of monolayer 1T'-WTe₂ nanoribbon. The theoretical estimation very much corroborates the localization of the charge distribution, visualizing the existence of three different edge modes (noted as A, B, and C).

Figure R8. Figures taken from published literatures. **a.** The dI/dV spectra measured by scanning tunneling spectroscopy across the step edge of 1T'-WTe₂. The conductance distribution represents the localized density of states of the edge, where the 1D helical edge state is confined [*Nat. Phys.* **13** 683-687 (2017)]. **b.** Theoretical calculation of the

wavefunction amplitude for the monolayer 1T'-WTe₂ nanoribbon, based on the tight binding model. Strong charge concentration at the edge is shown [*Adv. Mater.* **28** 4845-4851 (2016)].

Given the above two considerations, we see that the majority of spin-polarized electrons would originate from the edge with a small background signal from the bulk. These discussions are now included in the revised main text.

Edited contents

Added sentences

- Main text:

(Main text, page 5, line 1)

Here we note that although the remote pump area of the interior bulk is much larger than the 1D edge, the majority of spin-polarized electrons would originate from the edge with a small bulk background. This is consistent with the Ref. [19,20], where the bulk is lack of distinct helical states and the density of states of the edge is significantly larger than the bulk.

General remarks and comment of Reviewer 3:

This manuscript reports the valley Hall effect of spin-polarized electrons in MoS₂/WTe₂. Experimental signatures from both the excitonic PL intensity variation and Kerr-rotation are indicated, which unambiguously mark the presence of unipolar valley Hall transport in monolayer MoS₂. The anomalous Hall effect with a topological origin in two-dimensional layered semiconductors is an attracting topic, and several experiments performed in monolayer transition metal dichalcogenides have indicated such phenomena in recent years. The valley Hall effect of electrons in this manuscript can serve as an important complement to the previous experiments related to excitons. The experimental data shown in the manuscript is beautiful and clear. However there are still several points need the authors to clarify before I can recommend the publication:

Response:

We appreciate the time that Reviewer 3 took to read our paper. Reviewer 3 raised a concern about the effect of the small spin splitting of the MoS₂ conduction on the valley and spin polarization. He/she also has requested a clear explanation about the gate voltage dependence of the valley Hall mobility. In fact, this comment [comment 3-2] is in line with the one from Reviewer 2's comment [comment 2-5].

Note the term “non-local pump” is replaced with “remote pump” in the revised manuscript according to the comments of other reviewer.

We wish to point out that we have made a device label following Reviewer 1's advice; in the revised manuscript, device # 1 means the device we used for the most of experiments described in the main text, and device #2 indicates the auxiliary device shown in Supplementary Fig. 4. This is because Reviewer 1 suggested to provide data from another device to compare the injection efficiency of spin-polarized electrons. The additional experiments performed using the above devices are supplemented in the revised manuscript. Accordingly, we have labeled the device number in the revised manuscript.

Figure R90. Optical microscope images of devices used in the experiments. a. Device #1 is used for most of the experiments in the main text. b. Device #2 and its associated properties are added in Supplementary Fig. 4 and 7.

Below we present our point-by-point response to Reviewer 3's comments.

Comments 3-1:

(1) The sign of the electron Berry curvature, which determines the Hall velocity direction under a given in-plane electric field, is related to the valley but not the spin. The existence of a valley Hall effect then implies that the excited spin-polarized electrons are also valley polarized. A transport measurement has indicated that in MoS₂ the conduction band spin splitting is extremely small (~0.8 meV, see Nat. Commun. 8, 1938 (2017)), thus it's hard to say that the electron's spin and valley indices are locked. So the authors should discuss how the spin polarization in WTe₂ is converted to valley polarization in MoS₂. Meanwhile from the experimental observation one can get the sign of the Berry curvature thus the electron valley index. Is the obtained spin-valley relation consistent with those in the other papers?

Response 3-1:

As Reviewer 3 pointed out, the spin splitting of the monolayer MoS₂ conduction band is much smaller than the valence band. Despite such a weak correlation between the spin and valley, we have observed the spin polarization injection efficiency of 11.2 % for device #1 and 9.8 % for device #2; please see our **Response 1-2** to the Reviewer 1's comment for the calculation of injection efficiency. We would like to note that it is experimentally challenging to address the exact conversion mechanism from the electron spin polarization in WTe₂ to the valley polarization in MoS₂. If the valley polarization is not originated from the injection of spin-polarized electrons due to the small spin splitting, our experiment results may purely arise from the spin-polarized electron regardless of the valley degree of freedom. However, we wish to emphasize that the transverse Hall transport observed in our experiments is a strong signature of the valley Hall effect rather than the spin Hall effect, due to the following reasons.

First, the Kerr rotation signal exhibit nanosecond-long decaying transients, denoting a slow dynamic process. On the other hand, the spin polarization of the MoS₂ conduction band electron exhibits a lifetime far shorter than the nanosecond at 77 K [*Nat. Phys.* **11**, 830-834 (2015)] because of the highly efficient Elliot-Yafet spin relaxation of the small spin splitting. Therefore, the long-lasting Kerr rotation signal represents the valley polarization, and thus the transverse Hall transport detected by the same Kerr rotation signature is the evidence of the valley Hall effect. Secondly, the anomalous Hall conductivity calculated from the experiment results matches very well with the valley Hall conductivity. For an electron-doped system (such as our case), the spin Hall conductivity is about λ/Δ of the valley Hall effect, where 2λ is valence band spin splitting and Δ is the bandgap [*Phys. Rev. Lett.* **108**, 196802 (2012)]. Considering an order-of-magnitude difference between λ and Δ of the monolayer MoS₂, the valley Hall effect dominates the spin Hall effect in the total anomalous Hall transport.

Additional concern that Reviewer 3 raised is to check the consistency of the spin-valley relation compared to the previous reported studies. The spin-valley relation in our experiments can be obtained by the direction of the transverse valley Hall transport based on the helicity of the remote pump. We found the left-moving valley polarization is consistent with the prior studies for valley Hall transport direction [*Science* **344** 1498-1492 (2014); *Nat. Nanotechnol.* **11** 421-425 (2016)] as well as for the spin-valley relation [*Phys. Rev. Lett.* **108** 196802 (2012); *Nat. Mater.* **16** 1193-1197 (2017)].

The above discussion is newly added to the revised manuscript to elaborate the effect of the small spin splitting of the conduction band.

Edited contents

Added sentence

- Main text:

(Main text, page 8, line 10)

In fact, the spin splitting of the monolayer MoS₂ conduction band is much smaller than the valence band [35]. If the valley polarization is not originated from the injection of spin-polarized electrons due to the small spin splitting, our experiment results may purely arise from the spin-polarized electron regardless of the valley degree of freedom. However, the transverse Hall transport observed in our experiments is a strong signature of the valley Hall effect rather than the spin Hall effect, due to the following reasons. First, the Kerr rotation signal exhibit ns-long decaying transients, denoting a far slower dynamic process when compared with the spin lifetime of the MoS₂ conduction band at 77 K [36], because of the highly efficient Elliot-Yafet spin relaxation of the small spin splitting. Therefore, the long-lasting Kerr rotation signal represents the valley polarization. Secondly, the anomalous Hall conductivity calculated from the experiment results matches very well with the valley Hall conductivity. For an electron-doped system (such as our case), the spin Hall conductivity is about λ/Δ of the valley Hall effect, where 2λ is valance band spin splitting and Δ is the bandgap [1]. Considering an order-of-magnitude difference between λ and Δ of the monolayer MoS₂, the valley Hall effect dominates the spin Hall effect in the total anomalous Hall transport.

Comments 3-2:

(2) More details are needed to understand the relation between the rising time and the gate voltage in FIG. 4b. The magnitude of the valley Hall velocity is given by the product of the Berry curvature and the electric field, whereas the Hall conductivity also depends on the electron density. Is the different rising time here caused only by the Berry curvature's variation with the electron energy or wave vector k (given in Supplementary Note 7)? If so, what is the relation between V_G and the Berry curvature? On the other hands, does the electron density play any role here?

Response 3-2:

Before replying to this comment, we have realized that Reviewer 2 and Reviewer 3 raises almost the same concern (see **Response 2-5 and Response 3-2**). So, we simply address the same explanation here.

Both Reviewer 2 and 3 raised a concern about the role of V_G on the valley Hall velocity. In our experimental regime, the different rising time likely arises from the Berry curvature variation rather than the electrostatically induced uniform charge carriers.

For the valley Hall mobility, we wish to note that it does not exactly mean that mobility is proportional to the amount of injected electrons. Instead, changing V_G , i.e. increase or decrease of the total electron population, leads to the changes of valley Hall mobility by rendering more or less electrons to be affected by a larger or smaller Berry curvature distribution in the momentum space [*Phys. Rev. Lett.* **99** 236809 (2007)].

Just to illustrate the main point, we reiterate once again that the intrinsic valley Hall velocity \mathbf{v}_\perp depends on the Berry curvature $\Omega(\mathbf{k})$ and the electric field \mathbf{E} applied to the electron in the band [*Rev. Mod. Phys.* **82** 1539-1592 (2010), *Phys. Rev. B* **75** 045315 (2007)],

$$\mathbf{v}_\perp = \frac{1}{\hbar} \Omega(\mathbf{k}) \times e\mathbf{E} .$$

Here, the Berry curvature is an intrinsic character determined by the lattice symmetry of the involved atomic structure. As long as the longitudinal electric field is constant, which for most of the cases are done by applying DC source-drain potential gradient [*Science* **344** 1489-1492 (2014); *Nat. Nanotechnol.* **11** 421-425 (2016); *Nat. Commun.* **10** 611 (2019)], the transverse valley Hall velocity would not be changed. This is exactly the same as our case.

Figure R11. Schematic diagram describing the effect of the longitudinal field (\vec{E}_y) on the electron distribution. K point of the momentum space is expressed as a red dashed line, and the bottom of the conduction band near the Fermi surface E_F is marked as a black dashed line. As V_G increases, the increased amount of electron injection and the higher Fermi surface lead the change of electron distribution after thermalization from (i) to (ii). Note that Berry curvature is concentrated near at K point. More electrons experience a larger Berry curvature in case (ii) compared to the case (i).

In our experiment, changing the electron density by V_G controls the electron distribution in the MoS₂ conduction band. The applied longitudinal electric field \vec{E}_y makes the Fermi surface being tilted in momentum space, as schematically shown in Fig. R11. After the thermalization and cooling are finished, the injected group of electrons fills the conduction band from the point marked as a black dashed line in Fig. R11. When V_G increases, the electron population at the K (or K') point increases, whose effect is explained as the band filling from (i) to (ii). This appears as a faster rising dynamic component in our time-resolved Kerr rotation because the non-zero Berry curvature is concentrated at the K and K' point of the band extrema in the momentum space, i.e. the tilted Fermi surface makes the injected electrons away from the K or K' point.

We thank both Reviewer 2 and Reviewer 3 to point out this important issue. We have included the above discussion in the revised manuscript as well as in the revised Supplementary Note 8.

Edited contents

Edited sentences

- Main text:
(Main text, page 6, line 22)

The intrinsic valley Hall velocity \mathbf{v}_\perp depends on the Berry curvature and the electric field \mathbf{E} applied to the electron in the band [3,32],

$$\mathbf{v}_\perp = \frac{1}{\hbar} \boldsymbol{\Omega}(\mathbf{k}) \times e\mathbf{E} . \quad (1)$$

Note the Berry curvature is an intrinsic character determined by the lattice symmetry of crystal solids. In our experiment, changing V_G controls the valley Hall velocity by controlling the electron population in the K and K' point of the MoS₂ conduction band, where the non-zero Berry curvature is locally concentrated (see Supplementary Note 8 for detailed explanation).

(Main text, page 7, line 12)

Based on the above interpretation, we compare the experimental data with the theoretical estimation by calculating the valley Hall conductivity (see Supplementary Note 7 for detailed calculations).

(Main text, page 8, line 6)

Compared to Fig. 4b, while the valley Hall velocity follows $v_{\text{VH}} \propto e^{-V_G}$, the ns-long decaying time monotonically increases, which is independent of the velocity of electron wavepacket. It represents that the intrinsic mechanism is the dominant origin in our case.

- Supplementary Note 7:

(Supplementary Information, page 13, line 13)

The valley Hall velocity v_{VH} can be expressed as equation (1) using the electric field and the Berry curvature [11]. Besides, we used the approach of calculating valley Hall conductivity σ_{VH} to estimate the valley Hall mobility for simpler calculation.

$$\sigma_{\text{VH}} = -\xi \frac{e^2}{\hbar} \sum_n \int \frac{d\mathbf{k}}{(2\pi)^2} f(\varepsilon_n(\mathbf{k})) \boldsymbol{\Omega}_C(\mathbf{k}), \quad (\text{S5})$$

where ξ is the antisymmetric tensor, n is the band index in the 2D limit, $f(\varepsilon_n(\mathbf{k}))$ is Fermi-Dirac distribution, and $\boldsymbol{\Omega}_C(\mathbf{k})$ is the Berry curvature derived in equations (S3) and (S4). Based on the Berry curvature and known parameters of the monolayer MoS₂ [10], the intrinsic contribution can be calculated. Then, the valley Hall transport mobility μ_{Hall} is derived as a function of ρ for the comparison with the experimental results.

Added sentences

- Supplementary Note 8:

(Supplementary Information, page 14)

Supplementary Note 8 – Gate voltage dependence of the valley Hall velocity

For the valley Hall mobility, we wish to note that it does not exactly mean that mobility is proportional to the amount of injected electrons. Instead, changing V_G , i.e. increase or decrease of the total electron population, leads to the changes of valley Hall mobility by rendering more or less electrons to be affected by a larger or smaller Berry curvature distribution in the momentum space.

Remind that the intrinsic valley Hall velocity \mathbf{v}_\perp depends on the Berry curvature $\Omega(\mathbf{k})$ and the electric field \mathbf{E} applied to the electron in the band as shown in equation (1). As long as the longitudinal electric field is constant, which for most of the cases are done by applying DC source-drain potential gradient [4,12,15], the transverse valley Hall velocity would not be changed. This is exactly the same as our case. In our experiment, changing the electron density by V_G controls the electron distribution in the MoS₂ conduction band. The applied longitudinal electric field \vec{E}_y makes the Fermi surface being tilted in momentum space, as schematically shown in Supplementary Fig. 13. After the thermalization and cooling are finished, the injected group of electrons fills the conduction band from the point marked as a black dashed line in Supplementary Fig. 13. When V_G increases, the electron population at the K (or K') point increases, whose effect is explained as the band filling from (i) to (ii). This appears as a faster rising dynamic component in our time-resolved Kerr rotation because the non-zero Berry curvature is concentrated at the K and K' point of the band extrema in the momentum space, i.e. the tilted Fermi surface makes the injected electrons away from the K or K' point.

Added figure

- Figure R11 is newly added to the supplementary information as Supplementary Fig. 13.

Reviewers' Comments:

Reviewer #1:

Remarks to the Author:

The revised manuscript is now suitable for publication.

Reviewer #2:

Remarks to the Author:

The authors have addressed carefully all the reviewer's comments in very detail. The questions I raised have also been clarified. Although some requests of new measurements have not been performed due probably to some practical reasons, it doesn't compromise the novelty and correctness of the experimental results. I believe that the comments raised by other reviewers have also been addressed satisfactorily. Therefore, I would like to recommend the revised version for publication in Nature Communications.

Reviewer #3:

Remarks to the Author:

In the reply, the authors have clarified the referees' concerns. I recommend the publication of the revised manuscript in Nat. Commun..